# Cycle-Contrast for Self-Supervised Video Representation Learning

**Quan Kong, Wenpeng Wei, Ziwei Deng, Tomoaki Yoshinaga, Tomokazu Murakami**
Lumada Data Science Lab. Hitachi, Ltd.
`quan.kong.xz@hitachi.com`

## Abstract

We present Cycle-Contrastive Learning (CCL), a novel self-supervised method for learning video representation. Following a nature that there is a belong and inclusion relation of video and its frames, CCL is designed to find correspondences across frames and videos considering the contrastive representation in their domains respectively. It is different from recent approaches that merely learn correspondences across frames or clips. In our method, the frame and video representations are learned from a single network based on an R3D architecture, with a shared non-linear transformation for embedding both frame and video features before the cycle-contrastive loss. We demonstrate that the video representation learned by CCL can be transferred well to downstream tasks of video understanding, outperforming previous methods in nearest neighbour retrieval and action recognition tasks on UCF101, HMDB51 and MMAct.

## 1 Introduction

Self-supervised learning has achieved unignorable development in the domains of natural language processing and computer vision recently. Different from supervised learning which depends on manually annotated labels of training data, self-supervised learning utilizes prior knowledge from training data itself to forge supervisory signals for learning representations, such as the consistency of disparate views of the same images[21, 6]. Learning in self-supervised fashion unleashes the potential of massive unlabeled data, and is expected to accelerate many industrial applications of deep learning where acquiring labeled data is expensive or difficult.

Comparing to natural language processing and image recognition where self-supervised learning leads to competitive results, video related tasks on the other hand, are still dominated by supervised learning. We suggest that the main reason is insufficient adoption of video specific prior knowledge for learning video representation. Most of existing works take temporal sequence ordering [20, 13, 36] or future frame prediction [8, 28, 18] as pre-text tasks for self-supervised learning of video representation, which assume that the nature of the correspondences across frames or clips could be generalized to represent a video. These methods give effective representations and decent results of downstream tasks, however we suggest that utilizing other nature characteristics of video can lead to different yet representative video representations. We notice that a video has two levels of representation, namely video and frame. Video representation is constructed by the continuous frame representations, thus the video representation is limited by the representation of frames, and reversely is the same. Inspired by it, we argue that good video representations have certain properties across both video and frame domains. Concretely, the representations of video and its frames are supposed to be close to each other across video and frame domains and distant to all the other videos and frames in corresponding domain, respectively. To make use of this nature, we propose Cycle-Contrastive Learning (CCL), a self-supervised method based on both cycle-consistency between video and its frames, and contrastive representations in each domain itself, in order to learning representations with

the above desired properties. To verify that our proposed method can learn a good representation and can be transferred to downstream tasks by fine-tuning, we show a competitive result of CCL on two tasks related to nearest neighbour retrieval and action recognition, demonstrating that CCL can learn a general representation and significantly close the gap between the unsupervised and supervised video representation.

The contributions of this paper can be concluded as : (i) We argue that video representation is structured over two domains, video and frame, and a good video representation is supposed to be closed across both domains yet distant to all the other videos and frames in corresponding domain, respectively. (ii) We design cycle-contrastive loss to learn video representation with the above desired properties, and our experiments suggest that learned representations lead to decent results of downstream tasks.

## 2   Related Works

**Self-supervised learning on video understanding.** Self-supervised learning methods have been proposed to learn general video representations from unlabeled data in various works [4, 5, 3, 36]. Kim et al. [13] and Xu et al. [36] proposed to learn the video representation from a temporal order prediction pre-text task. Wang et al. [30] proposed to capture information from both motion and appearance statistics along spatial and temporal dimensions to learn unlabeled video representation. Han et al. [8] proposed a pre-text by predicting the future representations of clip from a video based on the recent past. Benaim et al. [1] proposed a novel pre-text task by predicting the motion speed of objects in the video whether they move faster, at, or slower than their natural speed for learning the video representation. On the other hand, using external modality from video for self-supervised learning is also a typical way to learn a robust video representation. Sun et al. [26] proposed to use video and text sequences for cross-modal learning in the self-training phase. Xiao et al. [34] proposed to use slow and fast visual path networks that are deeply integrated with a faster audio path network to model vision and sound in a unified representation. Our work is only using vision modality for self-supervised learning. Different from the existed works with only vision modality, that almost focused on the correspondences across frames or clips, our work considers to find the correspondences across frame and video to learn the representation. Tschannen et al. [27] proposed to apply different pre-text tasks on frame/shot (augmentation consistency) and video (future shot prediction consistency) respectively. However, CCL focuses on using the nature of belong and inclusion cycle-consistency relation across the frame and video with contrastive representations.

**Contrastive learning.** Contrastive learning learns representations by comparing positive pairs and negative pairs using a distance-based contrastive loss, presented by Hadsell et al. [7] and derived to various methods [11, 14]. Extended from the instance discrimination task [33], He et al. [9] proposed Momentum Contrast(MoCo) to build dynamic dictionaries that supports a large and consistent dictionary for unsupervised visual representation learning. Sermanet et al. [24] firstly applied contrastive loss to learn the video representation by comparing different-looking and similar-looking frames from multiple simultaneous viewpoints. Sun et al. In our method, we use contrastive loss for video and frame, to make sure that they are both discriminative in each domain itself and acquire a mutual promotion by introducing it to the cycle-consistency checking progress.

**Cycle-consistency.** cycle-consistency has been widely applied to various task. Zhou et al.[37] utilized the cycle-consistency across instances of the same category as the supervisory signal to learn dense cross-instance correspondences. CycleGAN presented by Zhu et al. [38] enables image-to-image translation when the paired data is not available by using cycle-consistency between source and target image domains. In the video understanding tasks, Wang et al.[32] used cycle relations of corresponding image patches in different frames of a video to solve the self-supervised object tracking problem. Dwibedi et al. [3] introduced a self-supervised representation learning method for video synchronization named temporal cycle consistency (TCC). It utilizes cycle-consistency to find correspondences across time in multiple videos by matching the corresponding frames in the embedding space. Inspired by this work, our method further exploits the cycle-consistency to match the correspondences between video and its frames in the embedding space. Different from the idea of TCC by checking the cycle-consistency of $frame \Leftrightarrow frame$ across prior paired videos which focuses on frame-by-frame alignment task, cycle-consistency in our approach follows $video \Rightarrow frame \Rightarrow video$ which involves the information across video and frame to encourage the network to learn the video representation from the belong&inclusion relation of video and frame.

# 3 Cycle-Contrastive Learning

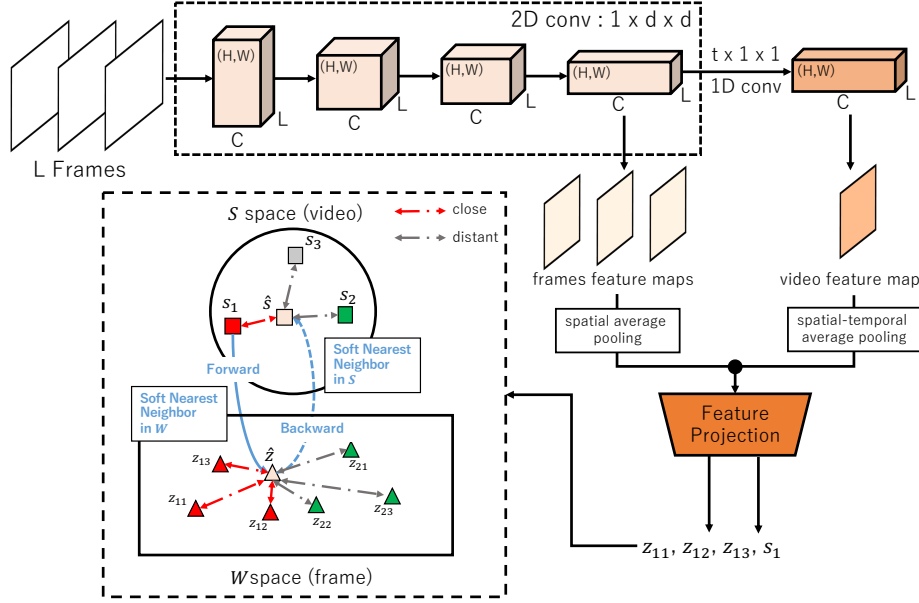

**Figure 1: Overview of Cycle-Contrastive Learning.** $L$ frames sampled data from video are input to a R(2+1)D like network. The outputs of the last 2D conv layer after spatial average pooling are used as frame features and the outputs of 1D conv layer after spatial-temporal average pooling are used as video features. Both of frame and video features are 512-dim vectors and then input to a feature projection module to be projected to 128-dim vectors. $z_{nm}$ is the projected feature of $m$-th frame from video $n$. $s_n$ means the $n$-th video's projected feature. $S$ is video embedding space, $W$ is frame embedding space. Cycle-Contrastive Learning is performed across the two spaces under a self-supervised manner by minimizing the similarity between the video and its frames and maximizing the similarity of them to all the other videos and frames in corresponding domain, respectively.

The core contribution of this work is a self-supervised approach to learn the correspondences across frame and video embedding spaces where the embeddings of video and its frames are close to each other and distant to all the other videos and frames respectively. We achieve such an objective by maximizing the number of video-and-frame pairs owning the cycle-consistency, additionally with a contrastive loss during cycle-consistency checking for both video and frame domains in a mini-batch. In order to facilitate learning such embedding spaces under the end-to-end manner using back-propagation. CCL can be divided into two phases: forward and backward cycle-contrast as shown in Fig. 1. We describe the details below.

## 3.1 Representation Projection

Let $V = \{v_1, v_2, ..., v_N\}$, given any video $v_i$ from $V$, the sampled frames from $v_i$ are defined as $F_i = \{f_{i1}, f_{i2}, ..., f_{iL}\}$ where $l$ is the sampled number of frames with a stride $\frac{M}{L}$, $M$ is the frame length of video sequence. $T = \{F_1, F_2, ...F_N\}$ denotes all the frames from $V$. The embeddings of video and frame are computed as $s_i, \{z_{iL}\} = \Psi(v_i, \theta)$ which are used for self-supervised learning, where $\Psi$ is the neural network with a non-linear transformation parameterized by $\theta$. $s_i$ and $\{z_{iL}\}$ are the embeddings for the input video and the set of its frames respectively. The non-linear transformation is a learnable MLP which is introduced between the cycle-contrastive loss and the representations of video and frame. We use the output from the last 2D conv layer after spatial average pooling as the frame representation without temporal downsampling and non-degenerate temporal convolutions (temporal kernel size $> 1$), which is a set of feature maps in correspondence to each input frame. And we use the output from 1D conv layer with non-degenerate temporal convolutions after spatial-temporal average pooling as the video representation. The representations of video and frame are not directly applied for self-supervised learning but projected to the same dimension latent space with the feature projection MLP. Assume that we are given a mini-batch including videos $V$ with size $N$

and their sequences $F$ with size $L \times N$. Their embeddings are computed as $S = \{s_1, s_2, ...s_N\}$ and $W = \{Z_1, Z_2, ..., Z_N\}$ where $Z_N = \{z_{N1}, z_{N2}, ..., z_{NL}\}$.

## 3.2 Cycle-Contrast

**Cycle-consistency** in our approach is to check the video cycle-consistency by leveraging the relation between video and its frames. If a given video embedding $s_i$ is cycle-consistent, we first determine its nearest neighbor $z_{nm}$ in the frame embedding space $W$, thus $z_{nm} = argmin_{z \in W} dis(s_i, z)$, where $dis(,)$ is a distance measurement between two vectors $s_i$ and $z$. According to the *inclusion* relation of the video and its frames, $z_{nm} \in Z_i$ is desired. This forward phase is defined as $video \Rightarrow frame$. We then use $z_{nm}$ to find its nearest neighbor $s_j$ in the video embedding space $S$ as $s_j = argmin_{s \in S} dis(z_{nm}, s)$. The video embedding $s_i$ is cycle-consistent only if $j = i$. Thus the video $s_i$ is cycle-back to itself according to the *belong* relation of the frame and its corresponding video. This backward phase is defined as $frame \Rightarrow video$. A good representation can be learned through maximizing the number of cycle-consistent samples for any videos. However, the representations learned by merely using cycle-consistency has no notion of how far from the embedding of the given video to the other videos in distance, and also the frames of the given video to the frames of the other videos. Therefore, we introduce a discriminative approach based on contrastive learning for the video and frame domains respectively.

**Contrastive learning** is applied for video and frame during the cycle-consistency checking. For forward phase $video \Rightarrow frame$, the nearest neighbor of $s_i$ in $W$ is $z_{nm}$, where $z_{nm} \in Z_i$. In addition, $z_{nm}$ is also desired to be disagree with any other frames $z \notin Z_i$. In other words, we treat $\{z_{i1}, z_{i2}, ..., z_{il}\}$ as the positive samples where $z_{nm}$ belongs, and the other $L \times (N-1)$ frames within a mini-batch as negative samples whose distance to $z_{nm}$ is maximized. We call this *forward cycle-contrast*. Similarly for backward phase $frame \Rightarrow video$, the nearest neighbor of $z_{nm}$ in $S$ is $s_j$, where $j = i$ and $s_i$ disagrees with any other videos. *Backward cycle-contrast* is to treat $s_i$ as the positive sample which $s_j$ should be, and the other $N - 1$ videos as negative samples whose distance to $s_j$ is maximized. To make forward and backward cycle-contrast differentiable, we use soft nearest neighbor following TCC[3] as introduced below.

## 3.3 Forward Cycle-Contrast

For the selected video sample $s_i$, we firstly compute the soft nearest neighbor $\hat{z}$ of $s_i$ in $W$ and then use it as the anchor point to be the positive pair $(\hat{z}, z_{ik})$ for each $z_{ik} \in Z_i$. We use softmax to define the soft nearest neighbor $\hat{z}$ of $s_i$ as:

$$\widehat{z} = \sum_{nm}^{L \times N} \alpha_{nm} z_{nm}, \; where \; \alpha_{nm} = \frac{exp(sim(s_i, z_{nm}))}{\sum_{nm}^{L \times N} exp(sim(s_i, z_{nm}))} \tag{1}$$

where $sim(u, v) = 1 - dis(u, v)$ means the cosine similarity between the two vectors $u$ and $v$, $dis(u, v) = 1 - u^\top v / \|u\|\|v\|$ is the cosine distance measurement. $\alpha_{mn}$ is the the similarity distribution which signifies the proximity between $s_i$ and each $z_{nm} \in W$ means $m$-th frame from video $n$. Then we use the positive pairs $(\hat{z}, z_{ik})$ to apply with contrastive loss in the frame domain $W$ which is defined similar to InfoNCE[22] as:

$$L_f = -\log \frac{exp(sim(\hat{z}, z_{ik})/\tau)}{\sum_{mn}^{L \times N} exp(sim(\hat{z}, z_{nm})/\tau)} \tag{2}$$

where $\tau$ denotes a temperature parameter. The forward cycle-contrastive loss $L_f$ means we encourage the network to make the soft nearest neighbor $\hat{z}$ of $s_i$ in $W$ close to the frame embedding in $Z_i$, and distant to the other frames in $W$ when minimizing $L_f$. The final forward cycle-contrastive loss is computed across all positive pairs in $(\hat{z}, z_{ik})$, $z_{ik} \in Z_i$.

## 3.4 Backward Cycle-Contrast

Similar to the forward phase, backward phase use $\hat{z}$ to find its soft nearest neighbor $\hat{s}$ in $S$. As the objective in backward is to cycle-back to the selected video sample $s_i$ and maximize the disagreement between $s_i$ to the other videos as well. The positive pair in this phase is $(\hat{s}, s_i)$ and the other $N - 1$

videos in $S$ are the negative samples of $\hat{s}$. The soft nearest neighbor $\hat{s}$ can be defined as:

$$\hat{s} = \sum_{j}^{N} \beta_j s_j, \; where \; \beta_j = \frac{exp(sim(\hat{z}, s_j))}{\sum_{k=1}^{N} exp(sim(\hat{z}, s_k))} \tag{3}$$

where the similarity vector $\beta$ defines the proximity between $\hat{z}$ and each $s_j \in S$. The backward cycle-contrastive loss is computed for the positive pair $(\hat{s}, s_i)$ defined as:

$$L_v = -\log \frac{exp(sim(\hat{s}, s_i)/\tau)}{\sum_{j}^{N} exp(sim(\hat{s}, s_j)/\tau)} \tag{4}$$

which signifies the cycle-back video sample is close to the selected video $s_i$ and distant to all the other videos in a mini-batch when minimizing $L_v$.

### 3.5 Penalization Term

The representations of different frames from the same video may suffer from the mode collapse problem if the learned embeddings of the same video's frames are similar to each other during cycle-contrastive learning. To avoid this, we need a penalization term to encourage the diversity of learned frame embeddings in the same video. Instead of maximizing a set of KL divergence between frames, we hereby introduce a penalization term without adding new learnable parameters and heavy computation. When $i$-th video is selected for cycle-consistency checkkng, we use the dot product of a matrix $R = [z_{1i}, z_{2i}, ..., z_{li}]$ and its transpose, subtracted by an identity matrix, which is similar to the way introduced in structured self-attentive sentence embedding [17] as:

$$P = ||RR^{\top} - I||_2^2 \tag{5}$$

and we minimize it together with the cycle-contrastive loss. Thus our final loss function to be minimized can be concluded as:

$$L = w_1 L_f + w_2 L_v + w_3 P \tag{6}$$

where $w_1, w_2, w_3$ are the balance parameters.

## 4 Experiment

In this section, we evaluate the effectiveness of our representation learning approach by using four datasets: Kinetics-400[12], UCF101[25], HMDB51[15] and MMAct[23] under standard evaluation protocols. The learned network backbones are evaluated via two tasks: nearest neighbor retrieval and action recognition.

### 4.1 Implementation Details

We constrain our experiments to a 3D ResNet. Table 1 provides the specifications of the network. It has $L = 8$ frames scaled to $128 \times 171$ and randomly cropped to the size $112 \times 112$ as the network input, which are sparsely sampled from the $M$ length clip with a temporal stride as $\frac{M}{L}$ with temporal jittering. To maintain the frame-level information before the temporal convolution for gathering video-level information, unlike typical R3D model that performs spatial-temporal downsampling in res blocks, we opt to not perform temporal downsampling and non-degenerate temporal convolutions from conv1 to res5 layer of which filters are essentially only 2D kernels with convolutional striding of $1 \times 2 \times 2$. We add one convolution layer as conv6 after res blocks for temporal convolution with a striding of $3 \times 1 \times 1$ of which the filters are 1D convolution kernels. We use the outputs from res5 with spatial average pooling and conv6 with spatial-temporal average pooling as the representations of each frame and video respectively. The representation of each frame and video is a 512-dimensional vector input to the feature projection module. The feature projection module is a 2-fc-layers MLP to project the representations of video and its frames to a 128-dimensional latent embedding space for computing cycle-contrastive loss in a mini-batch. The network is trained end-to-end with the transformation module. We evaluate our approach by using the representation of frame and video (512-d vector) rather than the projected embeddings. The temperature parameter $\tau$ is set to 1 in Eq.2 and Eq.4. Balance parameters $w_1, w_2$ and $w_3$ in Eq.6 are set to be 0.2, 0.2 and 0.4. The training was performed on 4 GPUs, taking $\sim 8$ days on Kinetics-400 and $\sim 0.5$ day on UCF101.

**Table 1: Network architecture considered in our experiments.** Convolutional residual blocks are shown in brackets, next to the number of times each block is repeated in the stack. The dimensions of kernels are denoted by $\{T \times H \times W, C\}$ for temporal, spatial height, width and channel sizes. The series of convolutions culminates with a global spatial-temporal pooling layer that yields a 512-dimensional feature vector as the video representation.

| layer name | output size | Our 3D ResNet |
|:---:|:---:|:---:|
| conv1 | $L \times 56 \times 56$ | $(1 \times 7 \times 7, 64)$, stride $1 \times 2 \times 2$ |
| res2 | $L \times 56 \times 56$ | $\begin{bmatrix} 1 \times 3 \times 3, 64 \\ 1 \times 3 \times 3, 64 \end{bmatrix} \times 2$ |
| res3 | $L \times 28 \times 28$ | $\begin{bmatrix} 1 \times 3 \times 3, 128 \\ 1 \times 3 \times 3, 128 \end{bmatrix} \times 2$ |
| res4 | $L \times 14 \times 14$ | $\begin{bmatrix} 1 \times 3 \times 3, 256 \\ 1 \times 3 \times 3, 256 \end{bmatrix} \times 2$ |
| res5 | $L \times 7 \times 7$ | $\begin{bmatrix} 1 \times 3 \times 3, 512 \\ 1 \times 3 \times 3, 512 \end{bmatrix} \times 2$ |
| conv6 | $L \times 7 \times 7$ | $(3 \times 1 \times 1, 512)$, stride $1 \times 1 \times 1$ |
| | $1 \times 1 \times 1$ | spatial-temporal global average pool |

## 4.2 Dataset

In order to evaluate the performance of our proposed method, we conduct the experiments based on UCF101 and HMDB51, which are widely used in the evaluation of video understanding tasks. UCF101 contains 13,320 videos and 101 action categories. HMDB51 contains 6,849 clips divided into 51 action categories. Both of them are $\tilde{2}5$ fps videos. We also evaluate our method on MMAct dataset contains more than 36k video clips wtih 30 fps for 20 distinct subjects with 37 action classes under 4 fixed surveillance camera views for further generalizability checking. Kinetics-400 is a large-scale video action dataset containing 400 human action classes, with at least 400 video clips with $\tilde{2}5$ fps for each action. We use it as the self-supervised pre-train dataset for action recognition task.

## 4.3 Effectiveness of Cycle-Contrast

The goal of cycle-contrast is to learn the correspondence that the video and its frames are in agreement with each other and disagreement with the other videos and frames. To verify that the learned representations satisfy the cycle-consistency and contrastive feature, we use the learned representations for 3 kinds of nearest neighbor retrieval tasks introduced as below. We use the test split1 of UCF101 to evaluate our self-supervised method.

**(A) frame$\Rightarrow$video.** The frames extracted from the test set are used to query all clips from the test set. The cosine distances of representations between the query frame and all clips in the test set are computed. When the matched clip including the query frame, appears in the Top-$k$ nearest neighbors of the query frame, it is considered to be correct.

**(B) video$\Rightarrow$frame.** The clips extracted from the test set are used to query all frames from the test set. When the frame belonging to the query clip, appears in the Top-$k$ nearest neighbors of the query clip, it is considered to be correct.

**(C) frame$\Rightarrow$frame, video$\Rightarrow$video.** Following the experiment settings in [35] and [2]. For video$\Rightarrow$video, 10 clips are extracted per video for both the train and test set. Clips from the test set are used to query the clips from the training set. When the class of a test clip appears in the classes of $k$ nearest training clips, it is considered to be correct. For frame$\Rightarrow$frame, we extract 10 frames per video which are sampled from each clip in 10 clips as queries and keys with the same retrieval procedure as video$\Rightarrow$video.

**Table 2:** Results of nearest neighbour retrieval on UCF101 of our proposal. $F$ and $V$ represent frame and video, the left side of $\Rightarrow$ is used as query.

| | $F \Rightarrow V$ | | | $V \Rightarrow F$ | | |
|---|---|---|---|---|---|---|
| Methods | Top1 | Top5 | Top10 | Top1 | Top5 | Top10 |
| MSE | 15.3 | 24.2 | 33.1 | 22.2 | 32.1 | 38.4 |
| CCL | 26.1 | 39.0 | 48.4 | 34.4 | 46.6 | 56.9 |

**Table 3:** Retrieval of frame- and video-level results on UCF101 compared with other methods.

| | Methods | Top-1 | Top-5 | Top-10 | Top-20 | Top-50 |
|---|---|---|---|---|---|---|
| | MSE | 13.1 | 20.2 | 23.4 | 28.6 | 36.2 |
| | Jigsaw [21]('16) | 19.7 | 28.5 | 33.5 | 40.0 | 49.4 |
| $F \Rightarrow F$ | OPN [16]('19) | 19.9 | 28.7 | 34.0 | 40.6 | 51.6 |
| | Buchler *et al.* [2]('18) | 25.7 | 36.2 | 42.2 | 49.2 | 59.5 |
| | **CCL(ours)** | **32.7** | **42.5** | **50.8** | **61.2** | **68.9** |
| $V \Rightarrow V$ | MSE | 10.0 | 19.4 | 26.8 | 33.1 | 40.1 |
| | COP[35]('19) | 14.1 | 30.3 | 40.0 | 51.1 | 66.5 |
| | SpeedNet[35]('20) | 13.0 | 28.1 | 37.5 | 49.5 | 65.0 |
| | **CCL(ours)** | **22.0** | **39.1** | **44.6** | **56.3** | **70.8** |

**Self-Training Phase.** We train our network by using CCL on UCF101 train split 1. The mini-batch is set to 48 videos and use the SGD optimizer with learning rate 0.0001. We divide the leaning rate every 20 epochs by 10 for a total of 100 epochs. Weight decay is set to 0.005.

We can check how the cycle-consistency is satisfied in the learned representations according to the nearest neighbour retrieval performance between the video and its frames, which are designed as task (A) and (B). The performance of task (C) can be used for checking how well the contrastive feature is learned in the representations.

Table 2 shows the Top-$k$ accuracy with $k$ varying for all task (A) and (B). The higher the accuracy means the better the cycle-consistency between video and frame is satisfied. We also report metrics on the Mean Squared Error (MSE) which is formulated by only minimizing $\sum \|s_i - z_{ik}\|^2$. It is a straightforward way to make the frame and video satisfy the belong and inclusion relations.

As we can see that CCL outperforms MSE by over $+10.0$ points on top-1 accuracy for these two tasks. It shows that the representation learned by CCL is satisfied well on cycle-consistency compared with MSE that is trained by simply closing the distance between video and its frames' representations, which can not ensure video and its frames are nearest neighbours to each other simultaneously.

CCL aims to learn a good video representation that is supposed to be closed across both video and frame domains from cycle-consistency, yet distant to all the other videos and frames in corresponding domain for owning the discriminativeness. We use task (C) of nearest neighbor retrieval for comparing the discriminativeness of learned representation with other self-supervised methods. The self-training phase of CCL is the same as described in section 4.3. Table 3 shows the task (C) Top-$k$ accuracy for different methods. The top row in Table 3 shows the frame-level retrieval. The bottom shows the video-level retrieval. In comparison with other self-supervised methods, our method provides $+7.0$ and $+7.9$ points higher Top-1 accuracy than Buchler *et al.*[2] and COP[35] on $F \Rightarrow F$ and $V \Rightarrow V$ respectively. It shows that the video representation learned by CCL is more discriminative.

## 4.4 Generalizability of Learned Representation

The main goal of unsupervised learning is to train a model that can be used for transferring to other supervised tasks. We use action recognition as the downstream task for evaluating the generalizability, thus we fine-tune our self-supervised network on three action recognition datasets UCF101, HMDB51 and MMAct.

**Table 4:** Comparison with other self-supervised video representation learning methods by **fine-tuning all layers**.

| Method(year) | Backbone | #param. | U. pretrain | UCF101 | HMDB51 |
|---|---|---|---|---|---|
| Shuffle&Learn [20]('16) | AlexNet | 58.3M | UCF | 50.2 | 18.1 |
| VGAN [29]('16) | AlexNet | - | UCF | 52.1 | - |
| Luo *et al.* [19]('17) | AlexNet | - | UCF | 53.0 | - |
| OPN [16]('17) | AlexNet | 8.6M | UCF | 56.3 | 22.1 |
| Buchler *et al.* [2]('18) | AlexNet | - | UCF | 58.6 | 25.0 |
| MAS [30]('19) | AlexNet | - | Kinetics | 61.2 | 33.4 |
| COP [36]('19) | R3D-10 | - | UCF | 64.9 | 29.5 |
| ST-puzzle [13]('19) | R3D-18 | - | Kinetics | 65.8 | 33.7 |
| **DPC** [8]('19) | **R3D-18** | 14.2M | **Kinetics** | **68.2** | **34.5** |
| SpeedNet [1]('20) | I3D | 25.0M | Kinetics | 66.7 | 43.7 |
| *ImageNet pre-trained* | R3D-18 | - | - | 60.3 | 30.7 |
| Random init. | R3D-18+1 | - | - | 44.7 | 19.4 |
| **CCL(ours)** | **R3D-18+1** | 12.1M | Kinetics | **69.4** | **37.8** |
| *Kinetics pre-trained* | R3D-18+1 | - | - | 85.2 | 57.2 |
| CBT [26]('19) | S3D | - | Kinetics | 79.5 | 44.6 |
| AVSlowFast [34]('20) | R3D-50 | 38.5M | Kinetics | 87.0 | 54.6 |

**Self-Training Phase.** We pre-train our network by using CCL on Kinetics-400 train split. The mini-batch is set to 48 videos and uses the SGD optimizer with learning rate 0.01. We divide the leaning rate every 20 epochs by 10 for a total 80 epochs. Weight decay is set to 0.0001.

**Fine-tune Phase.** A fully connected layer with softmax is appended on the top of our network. Only FC layer is randomly initialized and the other layers are initialized from the trained network under self-supervised manner. The hyperparameters and data pre-processing are the same as self-training phase. The network is fine-tuned end-to-end as other methods by 35 epochs for all test datasets. We use center crops of 10 clips uniformly sampled from the video and average these 10 clip predictions to obtain the video prediction for reporting top-1 accuracy.

**Results on UCF101&HMDB51.** We report the Top-1 classification accuracy (%) on test split 1 of UCF101 and HMDB51, and compare with other fine-tuning all layers results from existing self-supervised methods in Table 4. The backbone, number of parameters and the dataset used for unsupervised pre-train are also shown. For random init., we train the network from scratch with random initialization. ImageNet and Kinetics supervised pre-train initialization results are reported as reference. It can be observed that CCL is better than the previous methods and surpasses the ImageNet supervised pre-training on both datasets. Compared with DPC [8], CCL achieves an improvement with +1.2 and +3.3 points on UCF101 and HMDB51 respectively under the similar setting with less #param., and also superior to SpeedNet [1] on UCF with +2.7 points with half of number of parameters. In comparison with ST-puzzle and COP which both use 16 frames per clip for training, 8 frames version of CCL is superior to them with +3.6 and +4.5 points on UCF101. The motivation behind ST-puzzle and COP is to learn a temporal feature which may correctly represent the temporal order, where the discriminativeness between the different samples are out of scope in these methods. CBT [26] and AVSlowFast [34] reported a significant higher accuracy than the other methods with a larger backbone. However, these methods are using external modality in the self-training phase, while CCL is only using vision modality for self-supervised training.

Table 5 shows the Top-1 classification accuracy on test split 1 of UCF and HMDB51 under a linear classification protocol. In this setting, only FC layer are fine-tuned with the corresponding datasets, and the other layers are fixed with the self-supervised learning initialized network. CCL is higher than ShuffleLearn and 3D-RotNet with +25.6 and +4.4 points respectively, and slightly inferior to CBT that using multi-modal self-training and S3D as a backbone network. These results show that the learned video representation by CCL can be transferred well on action recognition task.

**Results on MMAct.** To further check the generalizability of our learned representation, we test our approach on a video dataset MMAct which provides 4 fixed surveillance camera views. It's

**Table 5:** Comparison with other self-supervised video representation learning methods under **linear classification protocol**.

| Method | Backbone | #param. | U. pretrain | UCF101 | HMDB51 |
|---|---|---|---|---|---|
| Shuffle&Learn [20] | AlexNet | 58.3M | Kinetics | 26.5 | 12.6 |
| 3D-RotNet [10] | R3D-18(full) | 33.6M | Kinetics | 47.7 | 24.8 |
| **CCL(ours)** | **R3D-18+1** | 12.1M | Kinetics | **52.1** | **27.8** |
| CBT [26] | S3D | - | Kinetics | 54.0 | 29.5 |
| AVSlowFast [34] | R3D-50 | 38.5M | Kinetics | 77.0 | 40.2 |

**Table 6:** Comparison results on MMAct.

| Method | Cross-Session | Cross-Subject |
|---|---|---|
| Random init. | 62.2 | 58.1 |
| TSN [31] | 69.2 | 64.4 |
| CCL(FC) | 59.1 | 54.5 |
| CCL(E2E) | 65.1 | 62.8 |

**Table 7:** Ablation on our cycle-contrastive loss. Top-1 accuracy for each method.

| Method | UCF101 | HMDB51 |
|---|---|---|
| MSE | 52.8 | 22.9 |
| with $L_v$ | 63.3 | 31.1 |
| $L_v + L_f$ | 67.6 | 35.2 |
| $L_v + L_f + P$ | 69.4 | 37.8 |

visual domain is slightly different from self-supervised pre-train dataset Kinetics, that the videos in Kinetics are almost captured from movies. We test our approach with two metrics: cross-session and cross-subject, following the same split of train and test in [23]. The network is initialized by CCL with Kinetics and then fine-tuned with MMAct.

Table 6 shows the f-measure of our approach and a VGG-16 based fully supervised method TSN [31] reported in [23]. TSN is pre-trained by ImageNet with only RGB stream. MMAct provides class-balanced samples with almost 1k videos of each category for avoiding the risk of over-fitting when training from scratch. The f-measure of CCL(FC) which only fine-tunes FC layer with fixed CCL initialized network is still lower than the random init one. We also test our approach by fine-tuning all layers end-to-end for CCL initialized network as CCL(E2E). The f-measure is improved by +3.9 and +4.7 points compared with random, which shows that the representation learned by CCL is still general and helpful for transferring.

## 4.5   Ablation Experiments

Table 7 shows the ablation study on cycle-contrastive loss as Eq.6. MSE is the baseline setting without consideration about the cycle-consistency and contrastive feature in the training phase. Competitive results can already be obtained by using the cycle-consistency and contrastive feature of video with $L_v$ loss. When taking the constrative feature of frame-level into consideration, a significant improvement of over +4.0 point is confirmed compared with only using $L_v$. By adding the penalization item as Eq.5 into the loss function, we acquired a further improvement on both datasets, which shows the effectiveness of penalizing the same video's frames that are similar to each other.

## 5   Conclusions

We proposed a new self-supervised video representation learning method by finding belong and inclusion relations of video and its frames through cycle-contrastive loss. Note that, the representation learned by CCL is still one nature of a good video representation owns. As a potential area of improvement, it is considerable to introduce both temporal feature e.g, frame order, and cycle-contrastive feature to a pre-text task for learning a more generalized video representation.

# 6 Broader Impact

The objective of this work is self-supervised learning of video representation. As a positive impact of this area is to provide a mechanism to acquire transferrable representation of video without manually annotation of training data. Various downstream tasks about video understanding based on machine learning can be benefit from a good video representation, such as action recognition/detection, sports action scoring and video recommendation system, by fine-tuning the self-supervise learned model to different tasks with less amount of annotated data. It will free the society from task specific data accumulation to focus more on the task design by utilizing a transferrable video representation.

However, any data driven learning system faces the risk of fairness problem caused by the biased distribution of the training data. Several supervised based researches already focus on this area. In this work, even though our method is designed to learn a video representation satisfies a nature of the correspondences across video and its frames without semantic information, the above risk is still possible occurred when transferring the representation to the other supervised task by fine-tuning with a biased train data. Therefore, we suggest to take the train data bias into consideration before the learning even in a self-supervised manner and combine with the solutions already used in the supervised tasks regarding with fairness problem when fine-tuning according to the real-world usage.

# 7 Funding Disclosure

This research was done at Hitachi, Ltd. without other relevant funding.

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
