[Reviews · NeurIPS 2020]

Review 1

Summary and Contributions: The paper presents a self-supervised method to train video encoders using a combination of the temporal cycle-consistency and fa (instance discrimination) losses. The proposed method results in improved downstream task performance on a number of action recognition benchmarks.

Strengths: 1. The presented method is quite general and combines two self-supervised strategies (temporal cycle-consistency and instance discrimination). The formulation is quite elegant as they convert the problem of aligning two sequence of embeddings to aligning between the video and frame embedding spaces to train an encoder for video tasks. 2. Good ablation studies showing the importance of the different losses. 3. Experiments showing downstream performance on multiple datasets.

Weaknesses: 1. It would be nice to look at training and validation curves of the different losses. Is the model able to bring all three losses down simultaneously? Are there any tricks related to weighing these losses? 2. Originally TCC aimed to focus on temporally fine-grained actions. It would be interesting to investigate the efficacy of the learned features on tasks beyond action recognition. Recognizing action segments on Epic Kitchens [1] or Breakfast dataset [2] would make a stronger case for this paper. Measuring phase classification accuracy on the Pouring dataset [3] is also an option. [1] https://epic-kitchens.github.io/2019 [2] https://serre-lab.clps.brown.edu/resource/breakfast-actions-dataset/ [3] Temporal Cycle-Consistency Learning. Debidatta Dwibedi, Yusuf Aytar, Jonathan Tompson, Pierre Sermanet, Andrew Zisserman

Correctness: The methods presented seem correct.

Clarity: The paper is well written but some parts need clarification: 1. Use of the word transformer in Figure 1 is confusing as the word might refer to Transformer architecture but in the implementation the transformer refers to temporal convolutions. 2. The authors claim they train a 3D ResNet but except the last layer the size of kernel in time axis is 1 that is most of the modules are 2D convolutions. Usage of 3D ResNet is confusing as a reader expects entire network to be 3D convolutions not just the last layer. 3. How are the 8 frames chosen for training? How are the frames and embeddings chosen for evaluation? What is the fps during train and test?

Relation to Prior Work: Related work is covered well.

Reproducibility: Yes

Additional Feedback: **** POST REBUTTAL **** I thank the authors for the rebuttal and answering my questions. Overall I like the paper as it pursues a new technique to learn features. They have experiments on a number of video datasets. It would be a stronger paper if they included comparisons to new papers such as SpeedNet. However in the rebuttal they state the reasons why it is not fair to directly compare with these papers (SpeedNet using a different architecture from them and CBT being trained with language supervision). But even without these particular comparisons, this paper has an interesting contribution that is using the cycle-consistency losses to learn video and frame -level features. As this framework is quite general and is easily applicable to different domains (time-series data or data with sets), I hope the community would benefit more from learning about this training methodology. For this reason, I retain my previous recommendation of accept.


Review 2

Summary and Contributions: This paper proposes to utilize a cycle-consistency of the semantics between video clips and the frames constituting them. Specifically, they enforce two learning objectives: 1) the forward objective to make sure that the representation of a clip is close to the representation of its constituting frame. 2) Another backward objective to ensure that the representation, in the clip space, for the nearest neighbor of a frame is the actual clip that contains this particular frame. The authors conducted experiments on UCF101, HMDB, Kinetics400 and MMAct with the tasks of nearest neighbor retrieval and action recognition.

Strengths: + The topic of self-supervised video representation learning is important and very relevant to the NeurIPS community. + The proposed cycle-constraint between frames and clips is interesting.

Weaknesses: The experiments presented in this is paper is insufficient in several ways. I will elaborate below. - I am not sure what the authors try to convey through Table 2. It seems they are comparing MSE and contrastive loss, for which it's already widely proven, by many previous work, to be true that contrastive loss is superior than MSE for this task. Instead, what I think the authors should actually compare to are baselines that can prove the effectiveness of the proposed cycle constraints. For example, I think one important baseline would be directly applying instance discrimination (for which MoCo is a particularly effective variant that can be used) on frames. Another important baseline would be temporal cycle consistency (TCC), such that we can see if the proposed cycle constraint is indeed more effective compared to cycles that only happen between frames (as is done in TCC). - L216-7, "Clips from the test set are used to query the clips from the training set.", why not query in the test set? - Table 3, the more recent and stronger baselines like DPC, CBT ("Learning Video Representations using Contrastive Bidirectional Transformer"), AVSlowFast ("Audiovisual SlowFast Networks for Video Recognition") are missing. - Same applies for Table 4, the entries are outdated and many stronger baselines are missing (for examples in my previous comment).

Correctness: The method is mostly correct, however I do have the following doubt: - It's not clear to me about the reason to retrieve a soft nearest neighbor. For example, for Eq. 2, instead of using \hat{Z} (obtained from Eq. 1) as the query, what's wrong with directly using s_i , the clip representation as the query? The same question applies to the backward cycle-contrast section as well. - Eq. 6, there is no weight factor balancing each term? Can I assume they are all 1?

Clarity: There are some typos and sentences that are hard to understand, I list some examples below: - L96-100, "We use the output from the last 2D conv ... average pooling as the video representation", these two sentences are hard to understand. - L115-8, "However, the representations ... to the frames of the other videos.", this sentence is hard to understand. - L122, "to be disagree" -> "to be disagreeing"

Relation to Prior Work: There are some important related work that are missing. I list some examples below: - Though in different way, the idea of utilizing the frame-clip hierarchy has been explored before in this paper "Self-Supervised Learning of Video-Induced Visual Invariances". This should at least be discussed. - The "Contrastive learning" section of Related Work has missed several representative papers like "Unsupervised Feature Learning via Non-Parametric Instance-level Discrimination", "A Simple Framework for Contrastive Learning of Visual Representations", which give an incomplete picture about the evolvement of this topic and the current state-of-the-arts. - The "Self-supervised learning on video understanding" section of Related Work has missed several representative papers like "Learning to See by Moving", "Unsupervised Learning of Visual Representations using Videos", "Slow and steady feature analysis: higher order temporal coherence in video", "Learning Video Representations using Contrastive Bidirectional Transformer", and "Audiovisual SlowFast Networks for Video Recognition".

Reproducibility: Yes

Additional Feedback: POST REBUTTAL After reading the author's rebuttal, I don't think my concerns in the initial reviews are properly addressed. For example, the authors did not provide any results on this instance discrimination baseline that I was suggesting in my reviews. In fact, the authors wrote in rebuttal that "DPC is a state-of-the-art approach by utilizing the idea of instance discrimination on frame level.", which is not true -- DPC learns self-supervised features through predicting futures. Also, the rebuttal did not address the problem of lacking comparisons to some of the recent video SSL methods (e.g. CBT). Because of all these, I will keep my reject rating as in the initial review. -------------------------------------------------------- Given the many weaknesses of the experiment section (i.e., lack of comparison to recent video self-supervised learning methods and proper ablation study to demonstrate the effectiveness of the proposed cycle constraints), I don't think this paper is ready for publication.


Review 3

Summary and Contributions: This paper learns self-supervised video representations using cycle consistency between the full video representation and individual frame representations. It formulates the problem in a contrastive learning framework, and evaluates the performance of learned representations on downstream video classification benchmarks (e.g. UCF101, HMDB51, etc.) and video/frame retrieval.

Strengths: + The idea of applying cycle consistency between the frames and full video representation is quite interesting and novel to the best of my knowledge. Video representation learning is relevant to the NeurIPS community + The empirical results show modest improvement over some recent video representation learning approaches such as DCP [7].

Weaknesses: - The presentation quality of the paper can be improved both in terms of technical formulation and overall flow. Please see clarity section for details. - From the presented results, it is hard to judge if retrieval performance is evaluated fairly. For instance many of the methods in table 4 are not evaluated in retrieval performance. The ones that are evaluated in table 3 either uses AlexNet, or uses R3D but pre-trained on UCF (not on Kinetics similar to the submission). - The penalisation term in section 3.5 is not adequately motivated. Some more explanation and insight would help here. Also there is no empirical ablation of this component. Does it bring any major performance increase?

Correctness: There are many errors in the way the method presented. I'm assuming that they are all typos mainly by looking at the results. Most of the methods that they use (cycle-consistency and contrastive losses) have established implementations available online. I would assume that they used these implementations in their methods.

Clarity: Clarity is the major issue in this paper. - in Section 3.1. there is a confusion between "l" (number of frames) and "l" as the variable iterating over latent frame representations {z_li}. I would suggest using capital L for count (similar to the others such as M, N, etc.) and "l" for iterating over samples. This would clarify this section a lot. Also $n \in N$ in line 104 is not necessary and incorrect as N is not a set. - The exp(dis(.,.)) in equations (1),(2),(3) and (4) should be exp(-dis(.,.)) as these exp(.) requires similarity inside not distance. - The indicator function in equations (2) and (4) is not needed as positive samples also should be in the denominator. Please see InfoNCE [21] for details. - Fig.1. uses the term "Transformer" for a 2 layer MLP. The term "Transformer" already has an established meaning, I would avoid using it this way. - Section 3.6 may not be necessary as it doesn't bring much new information on top of the relevant part in related work. - Technical presentation of the paper needs a comprehensive check. - Proofreading the paper one more time for language would also be helpful.

Relation to Prior Work: The relevant related works are discussed adequately.

Reproducibility: Yes

Additional Feedback: Update after the rebuttal: I read the other reviews and the rebuttal. The authors address many points that I asked in my review. In overall the results may not be significantly better than the state of the art but they are mostly competitive. Besides the core idea in the paper is interesting and mostly novel as far as I know. They also have a decent execution of the idea. I was a bit concerned about clarity of equations but they appear to resolve all these confusions. I think the proposed idea and their execution deserves some attention from the community although it may not be clearly outperforming the state of the art. Hence I'll update my review to marginally-above-accept.


Review 4

Summary and Contributions: This paper proposes a way of self-supervised video representation learning, termed cycle-contrastive learning (CCL). The method utilizes the cycle from videos to frames, then from frames back to videos as a constraint to learn effective video representations. The authors demonstrated the performance on three video datasets.

Strengths: The proposed cycle contrastive learning is new for video representation learning.

Weaknesses: 1. Paper writing needs improvement. For example, Figure 1 needs more detailed caption to make it self-contained. I read it multiple times to understand what is the cycle and what is the pipeline. Readers may have questions like which transformer architecture you used? How do you form the anchor/pos/neg samples? What does z11, z12, z13 mean? 2. What is the linear probe accuracy on Kinetics400? I mean, if you fix the backbone and only finetune the last fc layer, what accuracy will you get for Kinetcis400 dataset? This is an important metric to evaluate the quality of the learned representations. Right now, all your experiments are to initialize the model by pretrained weights and finetune end-to-end, it is hard to know whether the pre-training stage learns good feature. 3. Performance on several datasets are not comparable to state-of-the-art methods. For example, recent CVPR 2020 paper SpeedNet, https://arxiv.org/pdf/2004.06130.pdf, reports much higher performance on both UCF101 and HMDB51 dataset than the proposed method. Take UCF101 as an example, SpeedNet achieves 81.1% accuracy while this submission only obtains 68.3%. I'm not saying you need to beat the state-of-the-art, it is just the performance gap is quite large. I expect some discussions on this.

Correctness: Yes

Clarity: Mostly yes. As I mentioned in weakness, the method is not clear enough. Figures and tables are not self-contained

Relation to Prior Work: Yes

Reproducibility: Yes

Additional Feedback: Post-rebuttal: I have read the authors' rebuttal. I think it partially addressed my concerns. However, there are two important issues that are not addressed by the rebuttal, one is lack of comparisons to recent video self-supervised learning approaches (like CBT and SpeedNet), the other is lack of experimental results (such as linear evaluation on Kinetics400 dataset, which I would say is very important to evaluate the quality of learned representations). Hence, given the paper's current stage, I will keep my score as reject.

[Author Response · NeurIPS 2020]

**Related Works.** Thanks for the reviewers to point out the missing references in our initial content, which will be
added in our final version. Paper "Video-Induced Visual Invariances" focuses on applying different pre-text tasks on
frame/shot (rotation or frame level augmentation consistency) and video (future shot prediction consistency like DPC)
respectively without reporting the performance on temporal related benchmarks. However, CCL focuses on using the
nature of belong and inclusion cycle-consistency relation across the frame&video with contrastive representations in
each domain by introducing a differentiable soft nearest neighbour formulation to train a single network e2e.

**Method Details. As R1 pointed out**, our backhone is not a typical 3D ResNet due to the motivation of maintaining the
frame-level information before the temporal convolution for gathering video-level information described as L178-179.
We will change the name of 3D ResNet in our model to "2D+1D ResNet". **As comments from R2**, we use soft
nearest neighbour due to that nearest neighbour searching in CCL is not a differentiable procedure. Soft formulation
for nearest-neighbours refer to [Neighbourhood Components Analysis] is a method by introducing a differentiable
cost function based on stochastic neighbour assignments, and we introduce this idea into our approach to enable the
cycle-contrast procedure to be e2e learnable. **As R3 pointed out**, the definition of $dis()$ is a cosine similarity shown
as L134, and we will change the name of $dis()$ to $sim()$ in our final version. Our implementation is to avoid the
query sample measuring the similarity with itself in the denominator, and we confirmed that indicator function used
in Eq.(2),(4) is not an appropriate way and will be removed in our final version. The index of $z$ will be modified
according to **the advice of R2**, $n \in N$ in L104 will also be removed, and section 3.6 will be moved to Related Works.
Regarding penalisation term $P$, we show an ablation on it in Table.6, and we acquired a further improvement by adding
$P$ with 1.8 points on UCF. **Further comments from R4**, we will add a detail description in the caption of Figure 1
with the architecture information of each component and the math symbol used in the Figure to make it self-contained.
**As R1,R2 pointed out**, the balance parameter of Eq.(6) are 0.4, 0.4 and 0.2 for $L_f$, $L_v$ and $P$ respectively for both
retrieval and action recognition settings. They will be added in the final version. We will also add a plot of our training
and validation curves for each loss in our final version. **As R1 and R3 suggested**, the term "Transformer" in Figure 1
will be replaced by "Feature Projector".

**Evaluation Details. As comments of R1**, 8 frames are sparsely sampled from $M$-frame length video at a temporal
stride as $\frac{M}{8}$ as described in L90-91 and L176-177. We will further clarify it in the caption of Table 1. The frame chosen
from test clips is based on the same way as the above. For UCF, HMDB51 and Kinetics, video fps are 25 and for
MMAct is 30. **As R1 and R2 pointed out**, TCC is not compared in our experiment. It is because we focus on the
downstream tasks under a fully unsupervised setting. TCC still needs prior knowledge about whether the clips are the
same action class or not. However, we agree with that it is interesting to check the results on other tasks, such as phase
classification, to further prove the effectiveness of CCL by comparing with TCC which focuses on only frame level
relation. We will add this experiment in our final version. **Further comments from R2**, similar as the settings in TCC
for checking the cycle-consistency effectiveness, in Table 2, we use MSE as our baseline which is a straightforward way
to make the frame and video satisfy the belong and inclusion relations. To further clarify the effectiveness of CCL and
avoid the confusion regarding Table 2, we will remove Table 2 and merge the MSE result into Table 3 and add a result
of MSE in loss ablation study in Table 6. The reason why we didn't compare with CBT and AVSlowFast in Table 3 and
4 is because these two methods are utilizing multi-modal information (CBT:video+language, AVSlowFast:video+audio)
for self-training, which is not a fair comparison with the other methods listed in Table 3 and 4 that only use the video
modality. However, **as R2 pointed out**, to give a complete picture of this topic, we will add the results of these two
methods in table 4 as reference in our final version. DPC is a state-of-the-art approach by utilizing the idea of instance
discrimination on frame level. In fact, we reported the result about the comparison with it in Table 4 and outperformed
it with +1.2 points on UCF and +3.3 points on HMDB51 under the similar setting. As the res block in CCL are 2D conv,
we achieved the above results with only **12.1M #param** that is lower than DPC with 14.2M. It is further confirmed that
the effectiveness of CCL. **As R3 pointed out**, the retrieval experiment follows the reported results as paper COP(2019)
and SpeedNet(2020). For further fair comparison, we will add the number of parameters of each model in Table 3 and 4.
The reason why we use the clip from the test set to query the clips in the training set is because we should follow the
same experiment setting as previous works' in Table 3. Moreover, using unseen data to query the seen data (initialized
database for querying) is also a real use case in the retrieval system such as EC service. **As R4 pointed out**, we reported
the e2e fine-tune result due to the fair comparison with the previous works that mostly done under this setting in Table 4.
Result about fixing the backbone and fine-tune the FC layers was reported only in Table 5 as CCL(FC). We evaluate our
approach under linear classification protocol on UCF and HMDB51 with 52.1 and 27.8 acc. respectively. It is higher
than ShuffleLearn(58.3M #param) and 3D-RotNet(33.6M #param) with +25.6 and +4.4 points respectively, and slightly
inferior to CBT that using multi-modal and S3D without providing the #param information according to CBT paper.
We did not refer to paper SpeedNet as it was released after our initial submission, which will be added in the final
version. SpeedNet achieves 81.1 by using a larger two-stream backbone S3D-G with 64 frames as input, while ours is
only 8 frames with RGB stream. They also report the result using a weaker backbone I3D (25M #param according to
paper [A New Model and the Kinetics Dataset]), showing a result about 66.7 which is lower than ours, and performed
worse than all metrics in retrieval task in Table 3 with S3D-G. We will add all the above results to our final submission.

[Meta-Review · NeurIPS 2020]

The paper received mixed reviews. The reviewers noted that the idea is very interesting and novel, but at the same time, some comparisons are missing. The rebuttal clarified some of the concerns from the reviewers. While the paper would be stronger with additional baselines, the rebuttal clearly identifies why these baselines may be unnecessary. Papers do not need to obtain state-of-the-art results to be published, especially if they pursue interesting and alternative lines of research. The AC recommends the paper for acceptance.